# Chronic Inflammation in PCOS: The Potential Benefits of Specialized Pro-Resolving Lipid Mediators (SPMs) in the Improvement of the Resolutive Response

**DOI:** 10.3390/ijms22010384

**Published:** 2020-12-31

**Authors:** Pedro-Antonio Regidor, Anna Mueller, Manuela Sailer, Fernando Gonzalez Santos, Jose Miguel Rizo, Fernando Moreno Egea

**Affiliations:** 1Exeltis Europe, Adalperostr. 84, 85737 Ismaning, Germany; 2Exeltis Germany GmbH, Adalperostr. 84, 85737 Ismaning, Germany; anna.mueller@exeltis.com (A.M.); manuela.sailer@exeltis.com (M.S.); 3Solutex SA. Avenida de la Transición Española 24, 28108 Alcobendas, Spain; fgsantos@solutexcorp.com (F.G.S.); fmoreno@solutexcorp.com (F.M.E.); 4Chemo OTC. Calle Manuel Pombo Angulo 28, 28050 Madrid, Spain; josemiguel.rizo@chemogroup.com

**Keywords:** PCOS, obesity, inflammation, specialized pro-resolving mediators (SPMs)

## Abstract

PCOS as the most common endocrine disorder of women in their reproductive age affects between 5–15% of the female population. Apart from its cardinal symptoms, like irregular and anovulatory cycles, hyperandrogenemia and a typical ultrasound feature of the ovary, obesity, and insulin resistance are often associated with the disease. Furthermore, PCOS represents a status of chronic inflammation with permanently elevated levels of inflammatory markers including IL-6 and IL-18, TNF-α, and CRP. Inflammation, as discovered only recently, consists of two processes occurring concomitantly: active initiation, involving “classical” mediators including prostaglandins and leukotrienes, and active resolution processes based on the action of so-called specialized pro-resolving mediators (SPMs). These novel lipid mediator molecules derive from the essential ω3-poly-unsaturated fatty acids (PUFAs) DHA and EPA and are synthesized via specific intermediates. The role and benefits of SPMs in chronic inflammatory diseases like obesity, atherosclerosis, and Diabetes mellitus has become a subject of intense research during the last years and since PCOS features several of these pathologies, this review aims at summarizing potential roles of SPMs in this disease and their putative use as novel therapeutics.

## 1. Introduction

Polycystic ovary syndrome (PCOS) is a wide-spread endocrine disorder affecting 5–15% of women in their reproductive age worldwide and is a frequent cause of infertility [1]. Diagnosis according to current international guidelines that apply the diagnostic “Rotterdam” criteria require the presence of at least two of the following features: hyperandrogenism, ovulatory dysfunction or/and polycystic ovary in ultrasonic scans, while other pathologies must be excluded [2]. PCOS is further characterized by an elevated LH/FSH ratio and often associated with obesity and insulin resistance leading to an hyperinsulinemic state in 80% of the obese women and 30–40% of the lean ones [3]. The etiology of the syndrome has not been clarified completely yet. Certain genetic patterns that impact for example synthesis, regulation and action of sex hormones and the insulin receptor, but also gestational factors like high maternal levels of androgens or AMH (anti-Müllerian hormone) seem to contribute to the onset of the disease [4,5,6]. Postnatal lifestyle factors like inadequate nutrition accompanied by a lack of physical exercise promote the development of the disease as they often result in obesity and disturbances of the glucose metabolism. In fact, hyperinsulinemia, independent of BMI, is a key contributor to PCOS pathogenesis [7,8] as it results in augmented androgen production in the adrenal cortex and follicles via stimulation of LH secretion, while concomitantly reducing SHBG (sex hormone binding globulin) synthesis [8]. Resulting elevated androgene levels interfere with follicular maturation [9] and may lead to the characteristic clinical manifestations like acne and hirsutism. Obesity on the other hand, may not only reinforce insulin resistance [10], but is a pathogenicity factor itself contributing to the endocrinologic disorder due to the altered hormone metabolism of adipocytes [11]. Furthermore, adipose tissue contributes to constant low-level systemic inflammation as will be described further below in this review. PCOS itself nowadays is considered a condition of chronic inflammation with elevated levels of leukocytes, pro-inflammatory cytokines, elevated white blood count and markers such as the C-reactive protein being detectable [12] and also affects women with a normal BMI [13].

## 2. Inflammation: Initiation and Active Resolution

Inflammatory processes are crucial for the survival of human beings and occur as a reaction to stimuli like injury or infections. Potential pathogens may enter a host due to a trauma, barrier breakage or microbial invasion and in order to regain the integrity of the organism, the invaders must be eliminated, removed and the functional state of affected tissue must be restored. During the acute inflammatory response, eicosanoid lipid mediator (LM) molecules that derive from the ω-6 poly-unsaturated fatty acid (PUFA) arachidonic acid (AA) are rapidly synthesized by cells of the innate immune system, such as granulocytes and macrophages that are recruited to the sited of the event [14]. For this step, the enzymes cyclooxagenases 1 and 2 (Cox-1/2) are utilized and prostanoids including prostaglandins (PG), leukotrienes (LTs) and thromboxanes (TXs) are synthesized from AA [15,16]. Additionally, mast cells and the classically activated M1 macrophages secrete different kinds of inflammatory cytokines including TNFα, IL-1 and Il-6—all of them being highly inflammatory substances that drive the generation of the classical inflammatory symptoms: redness, heat, swelling, pain, and loss of function [14,15,16]. Guided by the inflammatory eicosanoids and cytokines present at the site of infection, neutrophiles and monocytes leave the blood vessels and migrate into the tissue, thereby contributing to the formation of inflammatory exudates and progression of the inflammation process [14,15,16]. An effective initiation of the inflammatory response is essential for survival, but its self-limitation is equally important. Exceeding inflammation may lead to the phenomenon of a cytokine storm and subsequent life-threatening sepsis [17]. The failure to temporally limit inflammation results in chronic inflammatory diseases including cardiovascular and neurological disorders, auto-immune diseases, diabetes, and cancer [18,19]. For a long time, the resolution of inflammation had been considered a passive process with mere dissolution of inflammatory mediators and inflammation divided into initiation and resolution [20]. This view was changed by the discovery of distinct lipid mediator molecules that can “switch on” resolution processes in animal models and actively drive the resolution processes. These mediators comprise four different subgroups: the resolvins (Rvs), lipoxins (LXs), protectins (PDs) and maresins (Masr) and are depicted as SPMs: specialized pro-resolving mediators [21,22,23]. Resolvins, protectins, and maresins are synthesized from the ω-3 PUFAs EPA (Eicosapentanenoic acid) and DHA (docosahexaenoic acid) via certain intermediates (18-HpETE, 17-HpDHA and 14-HpDHA). Their biosynthetic pathways (see Figure 1 for an overview) involve certain lipoxygenases as well as the Cox-enzymes that take part both in the eicosanoid synthesis as well as in SPM production. The synthesis of LX in contrast starts from ω-6 PUFA AA [24,25]. Interestingly, Aspirin as an irreversible inhibitor of Cox-enzymes blocks prostanoid synthesis by modification of their catalytic domain. However, these enzymes remain active and are triggered to synthesize the SPM precursors 18-HpETE and 17-HpDHA. Resulting SPMS are called aspirin-triggered resolvins, -maresins, or protectins (AT-SPMs) and are potent mediators of resolution that are widely used in experimental approaches [21,24,26,27].

## 3. Significance of SPMs in the Resolution of Inflammatory Processes

Key processes of inflammation resolution involve termination of neutrophile recruitment, removal of the short-lived neutrophils and a switch in macrophage function to their anti-inflammatory pro-resolving state [28]. By now, many studies have elucidated crucial roles of SPMs in triggering key events of inflammation resolution that finally lead to tissue regeneration and homeostasis. Their actions are reviewed in detail by Serhan et al. (see for example references [24,25,26,29]). Briefly, SPMs were found to contribute essentially to cessation of neutrophil infiltration and subsequent macrophage-dependent phagocytosis of these neutrophils. They also mediate downregulation of pro-inflammatory chemokines and cytokines (TNF-α, IL-6, IL-8, IL-12), reduce the production of prostaglandins and platelet-activating factor (PAF) and activate anti-inflammatory signaling pathways. SPMs are also crucial for clearance of the site of infection by enhancing efferocytosis and phagocytosis, thus clearing the site from cell debris, apoptotic immune cells, and bacteria. Additionally, they promote tissue-regeneration and wound healing. It is important to note, that the activation of the resolution pathways is already initiated during the very first steps of inflammation, since the signaling pathways that promote the synthesis of pro-inflammatory prostaglandins are interlinked with the generation of the SPMs. Namely, prostaglandins PGE_2_ and PGD_2_ are required for the induction of type1-lipoxygenases necessary for the synthesis of the LXs, Rvs, and PD_2_ [30]. Thus, alfa signals omega during an inflammatory process and in contrast to prior perception, resolution of inflammation is an active process initiated concomitantly with the inflammatory process. [30]. Consequently, the inhibition of PG-synthesis impedes the class switching of lipid mediators and may delay resolution [31,32]. Based on the ever-growing knowledge on SPM biosynthetic pathways, their molecular structures, corresponding cell receptors and their functions Serhan et al. proposed distinct pathway options for the progression of an inflammatory response [22,24,25]. While ideally, the inflammatory process is resolved and the site of infection is cleared from pathogens and apoptotic cells, a failure to counterregulate the synthesis of pro-inflammatory molecules and their signaling pathways may lead to chronic inflammation.

Notably, the action of SPMs is also linked to the adaptive immune response. For example, LXA4 was shown to stimulate the action of natural killer cells [33]. On the other hand, D-series resolvins RvD1 and RvD2 have an impact on CD4+-T-cell differentiation and the lineage-specific secretion of interleukines. They also lead to decreased release of Il-2, IFN-γ, and TNF-α by CD8+-cells [34].

## 4. Chronic Inflammatory Diseases: Significance of SPMs

Insufficient resolution may result in chronic inflammation and SPMs play a crucial role in its suppression as was deduced from experiments with different animal models [24,25]. The tissue damage observed in Periodontitis, for example, can be attributed to the action of activated neutrophils that produce pro-inflammatory cytokines PGE2, LTB4 and LXA4. An infection with the causative oral pathogen *P. gingivalis* results in a strong inflammatory response leading to an upregulation of Cox-2, and recruitment of neutrophiles as demonstrated in Air Pouche mouse models. Supplementation of lipoxin LXA4 analogues was shown to reduce both processes. Within these mouse models also the development of systemic inflammation due to an infection with the oral pathogen has been demonstrated, as Cox-2 expression in further tissues like lung and heart is triggered by the infection. Utilizing a rabbit animal model that overexpresses lipoxygenase type I leading to LX4-levels that are up to 10 times higher than in wild type rabbits further demonstrated the importance of the SPM LX4 in this disease: Periodontitis-provoked bone-loss was reduced in transgenic animals and neutrophil infiltration was significantly reduced compared to wild type animals [35].

In a further experimental approach with transgenic mice, the role of 12/15 lipoxygenase and its biosynthesis products including SPMs LXA4, RvD1 and RvD2 for atherosclerosis was demonstrated. Atherosclerosis is initiated by an inflammatory reaction of the vascular endothelial cells that, vial endothelial secretion of pro-inflammatory cytokines, lead to recruitment of monocytes and leukocytes. Monocytes differentiate into macrophages that transform into foam cells clustered at the vessel walls. Resulting atheroma enlarge and narrow the vessel. Foam cells inside the plaques continue to secrete pro-inflammatory cytokines, reactive oxygen species and other mediators. Atherosclerosis is therefore considered a chronic inflammatory disease [36]. Interestingly, transgenic 12/15 lipoxygenase (12/15 LOX) mice seemed to be protected against atherosclerosis and this was attributed to the elevated expression levels of RvD1, PD1 and 17-HpDHA compared to wild type mice. It was further shown that LXA4, PD1 and RvD1 reduced the number of cytokines derived from endothelial cells as well as the amount of adhesion molecules (P-Selectin and VCAM-1) and additionally improved the uptake of apoptotic thymocytes. All these processes contributed to the anti-atherogenic influence supporting the notion that atherosclerosis is a result of vascular non-resolving inflammation. Interestingly, in these transgenic 12/15 LOX mice, a standard high-fat western diet disrupted the protective mechanisms [37] and proved the impact of nutrition on inflammation homeostasis. Further evidence for the resolutive action of SPMs in chronic inflammatory diseases were derived from a murine model of arthritis, in which resolvin RvD1 and its metabolic precursor 17-HpDHA were shown to reduce tissue damage and pain more efficiently than steroids [38].

The clinical picture of fibrosis may result from unresolved inflammation and epithelial or microvascular bruises. In animal models the role of exogenous aspirin-triggered LX4-analogues in the reduction of pulmonal fibrosis was demonstrated [39] and in further experiments both LXA4 and its analogue benzo-LXA4 were shown to reduce fibrosis in kidney [39,40,41]. Further trials have demonstrated that exogenously administered RvD1 can reduce the amount of pro-inflammatory mediator molecules that are formed after exposure to cigarette smoke and lung toxins [42] and a reduction in SPM levels has been associated the chronic lung diseases asthma and COPD in humans [43].

## 5. Obesity, Insulin Resistance and Chronic Inflammation

Adipose tissue also is a source of chronic inflammatory response that results from different mechanisms. Adipocytes themselves secrete pro-inflammatory factors like e.g., leptins, lipocalin, resistin, TNF-α, IL-6 and Il-1 entitled adipokines [44]. Furthermore, increased levels of free fatty acids are present in obese individuals. As these molecules represent primary ligands for Toll-like receptors, they are an important trigger for innate and adaptive immune response [45]. Furthermore, Macrophages constitute a considerable fraction of adipose tissue and are mainly representing the activated M1 phenotype [46]. Hence, they secrete pro-inflammatory cytokines like TNFα and IL6 that may also induce inflammatory responses in other tissues including the ovary [47,48]. Macrophage-derived pro-inflammatory signalling molecules also play important roles in the induction of insulin resistance in peripheral insulin sensitive cells [49].

Furthermore, pronounced adipocyte hypertrophy du to nutrient excess can lead to hypoxia resulting in characteristic features like oxidative stress, i.e., the generation of reactive oxygen species and necrosis of adipocytes. Both phenomena are potent triggers of an inflammatory response and lead to further recruitment and activation of macrophages and T helper cells [50,51]. In addition, B-cell development and activity is impaired in obese individuals and respective mice models compared to non-obese controls. This leads to the formation of pro-inflammatory B-cells without an antigenic trigger and also promotes insulin resistance [52]. These mechanisms further amplify the inflammatory response and represent linkages between innate and adaptive immune system in the context of obesity and insulin resistance. Detailed information on the role of individual adipokines in the promotion of the inflammatory process can be found in Ref. [44], for example.

Interestingly, apart from these interlinked signaling pathways, the relationship between metabolism and the immune response is also mirrored on a cellular level as there is evidence that preadipocytes may convert into macrophages in a suitable environment [53].

Summing up, obesity and inflammatory immune response are interlinked via several signaling pathways that also integrate routes towards obesity-related insulin resistance [49,54] and further frequent comorbidities like cardiometabolic diseases [55], fatty liver disease, asthma [56], and Diabetes Mellitus [49,54].

## 6. Chronic Inflammation in PCOS

In most cases PCOS is accompanied by obesity and insulin resistance (IR), affecting about 65-80% of all patients [57,58] and it is well-known that hyperinsulinemia, hyperandrogenism and obesity in this disease reinforce each other [8,11,58,59]. But PCOS also represents a state of chronic inflammation [60] resulting in part from excess visceral adipose tissue and its above-described pro-inflammatory mechanisms. Chronic low-level inflammation mirrored by elevated levels of pro-inflammatory cytokines, however, is also present in normal-weight PCOS-affected women but was mainly attributed to the fact that also normal-weight PCOS patients tend to have a surplus of visceral adipose tissue and intraperitoneal fat depots [61,62]. Recent studies have identified BMI and insulin resistance as main predictors for increased levels of CRP and white blood cells [12]. In addition, PCOS patients were shown to have a certain pro-inflammatory genotype characterized by alterations in the genes encoding for TNF-α, TNF receptor and IL-6 [63,64,65]. Furthermore, hyperandrogenaemia in PCOS not only contributes to enhanced visceral adiposity but was also shown to contribute to inflammatory processes. Excess androgens trigger the activation of MNC (mononuclear cells) which leads to enhanced production of reactive oxygen species (ROS) and activation of NFκB, which in turn, enhances the expression of the pro-inflammatory cytokines TNF-α, IL-6, and Il-1. TNF-α and IL-6 are known mediators of insulin resistance and hyperandrogenaemia was therefore found to have a negative impact on the insulin-mediated IRS-PI3K-Akt signaling pathway [66,67].

These correlations represent an important linkage of the metabolic features in PCOS and chronic inflammation. Furthermore, abnormal ovarian function has been associated with enhanced macrophage infiltration of the ovary and increased expression of TNF-α, IL-6 and IL-8 resulting in the activation of corresponding pro-inflammatory signaling pathways [48,68,69].

## 7. SPMs: Potential New Treatment Options for PCOS

Treatment of PCOS is restricted to alleviate its symptoms depending on the patient’s needs. Lifestyle interventions to reduce weight and improve the metabolic profile may result in significant improvements concerning fertility and metabolic parameters [70,71]. Clinical manifestations of hyperandrogenism like hirsutism and acne may be treated with antiandrogens and combined oral contraceptives and when it comes to the fertility wish, different forms of follicle stimulation, e.g., ovulation induction and various kinds of in vitro fertilization procedures, may be applied [2]. Also, insulin-sensitizing drugs like metformin and inositols have proven their usefulness in the treatment of PCOS as they impede the harmful effects of hyperinsulinemia. Successful restoration of ovulatory cycles resulting in improved pregnancy rates may thus be achieved [72,73]. Furthermore, treatment with myo-inositol was proven successful in different kinds of assisted reproductive therapies and improved oocyte quality [74].

However, the underlying chronic inflammatory processes have not been addressed systematically as therapeutic targets for PCOS. Some ongoing trials focus this aspect using ibuprofen [75] or a salicylate derivative [76] for therapy. Interestingly, poly-unsaturated fatty acids (PUFAs) have also entered the focus in the treatment of PCOS. The positive impact of properties of ω-3-PUFAs EPA and DHA on cardiovascular parameters and their anti-inflammatory potential have long been described [77,78]. Some studies have specifically examined the effect of PUFA-enriched diets on PCOS. During an in-vitro-experiment on granulosa cell cultures for example, treatment with EPA significantly increased expression of IGF-1 mRNA, while reducing Cox-2-mRNA expression compared to non-treated cell cultures. The effect was observed both in cell cultured derived from PCOS- affected women and healthy women [79]. Clinical trials with PCOS-affected women demonstrated beneficial effects on plasma testosterone levels, HOMA-index, and SHBG levels, but influence on blood glucose levels differed [80,81,82]. A summary is found in Table 1.

Albeit these results are promising, they could not be reproduced systematically [79].

As described in detail above, ω6-PUFA AA is a precursor for pro-inflammatory lipid eicosanoids (Prostaglandins [PGs], Thromboxanes [TXs] and Leukotrienes [LTs]) that are synthesized via the action of Cox-1 and Cox-2 (PGs) or 5-LOX (5-Lipoxygenase: synthesizing LTs). Eicosanoids are involved in modulating blood flow, endothelial permeability, neutrophil chemotaxis, and platelet aggregation, and therefore crucial during initial stages of inflammation (Figure 2).

Omega-3-PUFAs reduce the production of pro-inflammatory eicosanoids by replacing AA in phospholipid membranes and, in the case of EPA, competing for Cox and 5-Lox. Via this pathway, leukotrienes (LT) and E-series resolvins (RvEs) are formed, while the D-series resolvins (RvD), protectins (PDs) and maresins (Mar) derive from DHA via the action of lipoxygenases 5-LO and 12-LO (Figure 1). These SPMs are synthesized by macrophages, neutrophiles and partially by endothelial cells and are crucial for ending neutrophile influx, macrophage switching, and clearance of apoptotic cells thereby actively contribute to resolution of inflammation.

Modern western diet usually contains enhanced amounts of ω-6 PUFAs compared to ω-3-PUFAs and this non-beneficial composition has been shown to contribute to the development of cardiovascular diseases, cancer, inflammatory and autoimmune diseases [83]. This is partially explained by the fact that both kinds of PUFAs compete for the same enzymes and excess ω-6 PUFAs may shift the metabolism towards a pro-inflammatory state. Consequently, for chronic inflammatory states including cardiovascular disease [84], metabolic syndrome [83], diabetes mellitus [49], or the prevention and treatment of atherosclerosis [85], beneficial effects of ω3-PUFAs have been demonstrated. The protective effect originates from a reduction in pro-inflammatory eicosanoids and concomitant increase in pro-resolving mediators. However, it is the SPMs, the terminally active metabolites deriving from the PUFAs (see Figure 1) that act as resolutive mediators. In animal and in vitro models for adiposity, SPMs were shown to decrease pro-inflammatory cytokines (IL-6, TNF-α, IFN-γ), increase pro-resolution cytokine IL-10, counteract adipokine secretion and monocyte accumulation, promote M2 polarization of macrophages, and improve insulin sensitivity [86]. They work in nanomolar range, but the production depends on the presence and activity of Cox- and LOX enzymes as described above (see also Figure 1). In adipose tissue, elevated levels of pro-inflammatory eicosanoids compared to reduced SPM levels were found and it was assumed that the chronic low-grade inflammation results from failed resolution capacity [87] In a recent study, leukocytes from obese individuals with chronic low-grade inflammation were found to produced decreased levels of the 17-HDHA intermediate of the DHA-resolvin biosynthetic pathway and impaired synthesis of the D-series resolvins. This was a consequence of decreased amounts of 5-LOX, the enzyme responsible for synthesis of D-series resolvins from DHA. Hence, the administration of SPMs and their precursor metabolites may be more reliable for counteracting chronic inflammatory processes since this avoids potential enzymatic bottlenecks [85,86,88,89].

SPMs, including resolvin, maresins, and protectins, were shown to improve insulin sensitivity and glucose tolerance in mouse models [90,91,92,93] via stimulation of anti-inflammatory signaling pathways and reduction of pro-inflammatory macrophages. However, in a recent study that demonstrated a pronounced insulin-sensitizing effect of RvE1, the effect was not only credited to its pro-resolutive capacity, but also to rather pleiotropic functions that still needs to be elucidated in detail [89]. Similarly, glucoregulatory activity of protectin PDX different from its anti-inflammatory properties have been described in a further study [88].

Both obesity and insulin resistance represent fundamental features of PCOS that contribute to its pathogenesis and reinforce hyperandrogenemia. The underlying chronic inflammatory processes are also mirrored in PCOS and may be a promising target for a therapeutic approach.

Classical anti-inflammatory approaches aim at suppressing or blocking the action of pro-inflammatory mediators. Non-steroidal anti-inflammatory drugs (NSAIDs) for example inhibit Cox enzymes and prevent the conversion of AA into its prostanoids. Thus, the signs and course of inflammation are suppressed. However, NSAIDS are also known to deteriorate wound healing, increase the risk for osteoporosis and their use may lead to gastrointestinal bleeding. Furthermore, application of Cox-2 inhibitors increases cardiovascular risks. On the other hand, therapies targeting TNF signaling raise the risks for infections and lymphoma and non-melanoma skin cancer [9,20,94,95,96]. In addition, prostaglandins PGE2 and PGD2 are crucial for the initiation of resolution as described above and the inhibition of their biosynthesis may thus be counterproductive [25].

SPMs on the other hand show profound anti-inflammatory activity as discussed in detail above. They diminish the synthesis and action of pro-inflammatory mediators like LTs, PGs and PAF, stimulate the anti-inflammatory M2 phenotype of macrophages and increase the number of anti-inflammatory molecules like IL-10, limits recruitment of neutrophiles, triggers the macrophage switch to the anti-inflammatory M2 phenotype and increase their phagocytotic and efferocytotic action thus contributing to clearance of the site of inflammation [25]. In addition, they also stimulate tissue regeneration, which is essential for wound healing [97]. They are increasingly proposed for treatment of chronic inflammatory states like cardiovascular disease [84] or obesity and diabetes [49], but also their benefits in the prevention of tumor growth is a subject of intense research [98]. Most recently the putative role of SPMs and their precursors in the management of Covid-19 disease [99] were discussed A clinical pilot study including patients with coronary artery disease indeed demonstrated a beneficial effect of ω-3 supplementation on the lipid mediator profile: Pro-resolving mediators were restored in treated patients compared to non-treated individuals resulting in increased macrophage-driven uptake of clots. In a clinical trial focusing on patients with chronic pain a 4-week supplementation with SPM-enriched marine lipids resulted in significant reduction of pain and improved quality of life [100,101,102,103].

To date, there is no information available on the abundance and action of SPMs and their precursors in PCOS patients compared to healthy women. However, summing up the above-reviewed knowledge of SPMs and their precursors in key pathologic features of PCOS, it seems reasonable to expect an important role of these novel lipid mediators in the disease.

Hence, in preliminary clinical trials the abundance of SPMs and their precursor intermediates in PCOS-affected women compared to healthy individuals should be examined. Due to the importance of adipose tissue on inflammatory processes, the influence of the patient’s BMI on their lipid mediator profile should also be investigated. Later trials may focus on the dietary supplementation of SPMs and its effect on PCOS patients.

## 8. Conclusions

PCOS is a disease characterized by hyperandrogenemia and disturbed ovulatory function, but also by profound metabolic changes including insulin resistance and obesity and an underlying chronic low-grade inflammation. During recent years, the importance of active resolution processes in acute and chronic inflammatory processes has been uncovered and families of special pro-resolving lipid mediators (SPMs) have been identified and described. Increasingly, their role as putative therapeutic agents in inflammatory diseases like diabetes mellitus, atherosclerosis and many more is investigated. Since PCOS includes several features of chronic inflammation that contribute to its pathogenesis, SPMs might represent a novel therapeutic approach. Clinical investigations will be required to identify the putative role of the SPMs and their precursor molecules in this respect.

## Figures and Tables

**Figure 1 ijms-22-00384-f001:**
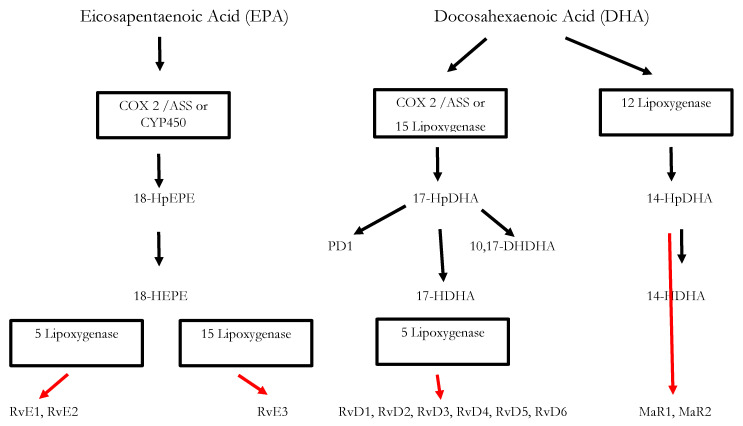
Biosynthesis of the SPMs Resolvins, Protectins and Maresins. EPA; Eicosapentaenoic Acid; 18-HpEPE, 17-HpDHA, 14-HPDHA: precursors of the SPMs during biosynthesis; Cox-1/2: Cyclooxygenases. ASS triggers biosynthesis of 18-HpEPE, 17-HpDHA intermediates via modification of Cox-enzymes. Maresins are produced by macrophages via a preliminary lipoxygenation step. Further lipoxygenases are required for SPM biosynthesis as depicted in Figure 1 [25].

**Figure 2 ijms-22-00384-f002:**
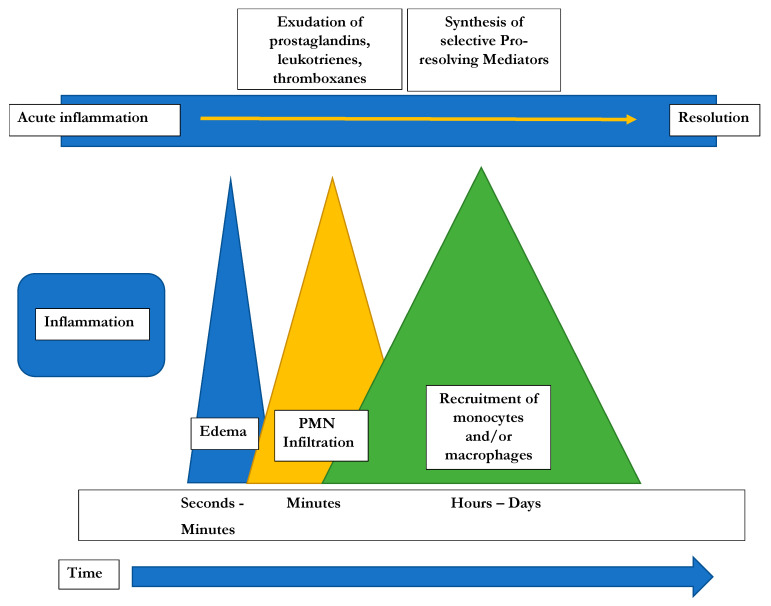
Time course of the potential development of inflammatory responses [22].

**Table 1 ijms-22-00384-t001:** Summary of studies on the effect of PUFAs on PCOS.

Object of Study	Design	Result	Reference
Influence of EPA on IGF-1 and Cox-3 expression in granulosa cells of PCOS women	In vitro cell culture of human granulosa cells from PCOS-affected women and healthy women. Exposition to EPA.	Significantly higher expression of IGF-1 m-RNA and lower expression of Cox-2 mRNA compared to non-treated control in both groups.	[79]
Effect of ω-3-PUFAs on metabolic and endocrine parameters	*N* = 104 women with PCOS. Effect of PUFA ω-3 supplementation on metabolic and endocrine parameters of a subgroup of *n* = 22 women;	Reduction of bioavailable plasma testosterone concentration. Modulation of lipid profile.	[80]
Effect of ω-3-PUFAs on obesity status and insulin resistance	*N* = 61 women with PCOS and BMI between 25 and 40 kg/m^2^; double-blind randomized trial. Daily supplementation with 1200mg ω3-PUFA or placebo over 8 weeks	No significant effects on weight, BMI or waist circumference, but significant improvement in blood glucose level and insulin resistance	[81]
Effect of ω-3-PUFAs on PCOS	*N* = 45 non-obese PCOS women, daily supplementation with 1500mg ω-3-PUFA for 6 months	Decrease in BMI, insulin resistance, but not in blood glucose levels; Decrease in serum LH levels and testosterone levels; increase in SHBG levels	[82]

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
