# Peer review of "Chronic Inflammation in PCOS: The Potential Benefits of Specialized Pro-Resolving Lipid Mediators (SPMs) in the Improvement of the Resolutive Response"

_ijms, 2020, doi:10.3390/ijms22010384_

Round 1

Reviewer 1 Report

This review consists mainly two parts. The first part summarized current understanding of inflammation and its resolution. Especially, the role of lipid mediators, SPMs, that resolve inflammation is well described. This part is well written though it is not so special to this article. 

The second part describes the inflammatory features of PCOS and proposes potential application of SPMs to PCOS treatment. This part should be original to this review manuscript. In that sense, this part is still less informative and should be described more in detail including studies with animal models, clinical trials with omega-3 PUFA supplementation, and clinical trials with SPMs themselves.

As for the clinical trials, the paragraph from line 251 and 260 should be expanded. For example, "some studies examined the effect of PUFA-enriched diets on PCOS. They demonstrated positive effects on metabolic parameters and insulin resistance but could not be reproduced systematically (78)." This sentence is so simple that we cannot understand how many studies have been done, how the outcomes were different, what are the possible reasons for the conflicts, and further possible approach. It would be nice if the previous studies were compared using a table. 

I guess clinical trials with SPMs on PCOS has not been reported yet. In that case, clinical studies with SPMS on other inflammatory diseases should be shortly summarized. Finaly, it would be nice if the authors discuss what issues to be solved before using SPMs on PCOS treatment.

Minor issues are as below

  1. Papers 37, 40 and 41 are not cited in the text.
  2. Figure 2 is not cited in an appropriate position.
  3. The citation in figure 5 and 6 is incorrect.
  4. There are several typos; line 45 concomittantly, line 90 aspirini-triggered, line 192 IL6 and Il1, line 243 akne. 

Author Response

We have implemented the changes asked for. This in the new manuscript format.

Reviewer 2 Report

The present literature review entitled "chronic inflammation in PCOS: the potential benefits of specialized pro-resolving lipid mediators (SPMs) in the improvement of the resolutive response" is well documented. However, I would like to stress the lack of care that was taken in formatting the manuscript. This has the effect of making the reading annoying. From the outset, the corresponding author is called "firstname Lastname". There is a lack of punctuation everywhere, there are too many spaces everywhere too. And the references have an extreme heterogeneity of format: I will cite as an example only the years, quoted at the beginning or end, in brackets or accompanied by a; , or . !

Finally, it drags in bold or underlined segments for no reason whatsoever.

Likewise, many acronyms are cited without the slightest initial explanation (BMI, LH / FSH, ...).

Redundant words are linked together (eg. Line 58: eliminated, removed) and even for enzymes, the typography varies: COX, Cox, CoX or even IL, Il !!!!

I told you: very annoying.

Following the same logic, many tipping errors remain, eg Figure 1 legend (18-HpEDE is indicated and not 18-HpETE).

Regarding the substance, as a general remark, I would indicate that it seems to me necessary to recast paragraphs 5, 6 and 7 because there are many unnecessary repetitions and a need sometimes to go back to the 5 when reading the 7 ...

Moreover, as the title augurs a work on the PCOS, and as the general reviews on the PMS being numerous and regular, it seems to me that paragraphs 2, 3 and 4 should be summarized.

In more detail,

 the sentence line 137-138 does not make sense.

Paragraph 7 is the worst worded:

line 250: oocyte

line 259-260: sentence piece not deleted?

Line 266: "... 5-LO" add: activity

line 272;

contribute instead of contributing

move up the sentence (line 282-283) to line 280

remove Fig 6 without interest

Author Response

We have introduced in the new manuscript the changes that were expected.

Round 2

Reviewer 1 Report

The manuscript has been well revised.